# Liver X Receptors Regulate Cholesterol Metabolism and Immunity in Hepatic Nonparenchymal Cells

**DOI:** 10.3390/ijms20205045

**Published:** 2019-10-11

**Authors:** Kaori Endo-Umeda, Makoto Makishima

**Affiliations:** Division of Biochemistry, Department of Biomedical Sciences, Nihon University School of Medicine, 30-1 Oyaguchi-kamicho, Itabashi-ku, Tokyo 173-8610, Japan; umeda.kaori@nihon-u.ac.jp

**Keywords:** cholesterol, liver X receptor, hepatic nonparenchymal cell, Kupffer cell, macrophage, NAFLD, NASH, hepatic fibrosis

## Abstract

Excess dietary cholesterol intake and the dysregulation of cholesterol metabolism are associated with the pathogenesis and progression of nonalcoholic fatty liver disease, nonalcoholic steatohepatitis, and fibrosis. Hepatic accumulation of free cholesterol induces activation of nonparenchymal cells, including Kupffer cells, macrophages, and hepatic stellate cells, which leads to persistent inflammation and fibrosis. The nuclear receptors liver X receptor α (LXRα) and LXRβ act as negative regulators of cholesterol metabolism through the induction of hepatocyte cholesterol catabolism, excretion, and the reverse cholesterol transport pathway. Additionally, LXRs exert an anti-inflammatory effect in immune cell types, such as macrophages. LXR activation suppresses acute hepatic inflammation that is mediated by Kupffer cells/macrophages. Acute liver injury, diet-induced steatohepatitis, and fibrosis are exacerbated by significant hepatic cholesterol accumulation and inflammation in LXR-deficient mice. Therefore, LXRs regulate hepatic lipid metabolism and immunity and they are potential therapeutic targets in the treatment of hepatic inflammation that is associated with cholesterol accumulation.

## 1. Introduction

The liver plays an essential role in the metabolism of nutrients, such as proteins, lipids and carbohydrates, and xenobiotics. Biosynthesis, catabolism, and efflux pathways in hepatocytes tightly regulate hepatic cholesterol homeostasis [1], and impaired cholesterol balance can lead to atherosclerosis and nonalcoholic fatty liver disease (NAFLD) [2]. Cholesterol accumulation induces activation of hepatic nonparenchymal cells, including Kupffer cells and stellate cells, and accelerates nonalcoholic steatohepatitis (NASH) and fibrosis. The nuclear receptors liver X receptor α (LXRα; also called NR1H3) and LXRβ (NR1H2) are transcription factors that regulate the expression of key genes that are involved in lipid and cholesterol metabolism [3]. LXRs belong to the NR1H nuclear receptor subfamily along with the bile acid receptor farnesoid X receptor (NR1H4) and they are closely related to the NR1I subfamily, which contains vitamin D receptor (NR1I1), pregnane X receptor (NR1I2), and constitutive androstane receptor (NR1I3). These NR1H and NR1I subfamily receptors regulate bile acid metabolism by sensing the metabolic environment [4]. Nuclear receptors have a structure that is comprised of an activation function 1 domain, a DNA-binding domain with a C4-type zinc finger structure, a hinge region, and a ligand-binding domain that contains an activation function 2. Ligand binding induces a conformation change in the receptor, which leads to the dissociation of a corepressor complex and recruitment of a coactivator complex and allows the receptor to induce the transcription of specific target genes. Nuclear receptors, including LXRs, also exhibit transrepression effects. LXRs are also expressed in hepatic nonparenchymal cells, including leukocytes, hepatic stellate cells (HSCs), and liver sinusoidal endothelial cells (LSECs), and they regulate immunity and inflammation, contributing to counter-regulation of hepatic inflammation, diet-induced NASH, and fibrosis. Thus, LXRs play important roles in both hepatocytes and nonparenchymal cells. The liver plays important roles in both lipid metabolism and innate immunity as a gateway for dietary signals, and LXRs are suggested to have a gatekeeper function in the liver. Although synthetic LXR agonists have been developed, their clinical application is limited by adverse effects, such as hypertriglyceridemia and neuropsychiatric symptoms [5,6]. Function-selective LXR agonists might have promising therapeutic potential for liver diseases by regulating lipid metabolism and immune responses. Here, we review the role of LXRs in hepatic nonparenchymal cell function and the pathogenesis of liver diseases.

## 2. Hepatic Nonparenchymal Cells

A variety of cell types are present in the liver: 70–80% are parenchymal cells, hepatocytes, and cholangiocytes, and 20–30% are nonparenchymal cells, including leukocytes, HSCs, and LSECs [7,8]. As the liver samples bacterial components and xenobiotics, together with nutrients, from the portal vein, immune cells are present and have a patrolling function in the sinusoids. Hepatic immune cells principally consists of resident Kupffer cells, bone marrow-derived macrophages (BMDMs), natural killer (NK) cells, natural killer T (NKT) cells, T cells, and B cells. These cells play essential roles in the regulation of metabolism, host defense, anti-tumor immunity, and regeneration as a functional network in combination with hepatocytes [9].

### 2.1. Two Types of Macrophages: Resident Kupffer Cells and Bone Marrow-Derived Macrophages (BMDMs)

In the steady-state condition, there are two types of macrophages in mouse liver: resident Kupffer cells and BMDMs. Fate mapping studies show that resident Kupffer cells originate from erythro-myeloid progenitor cells expressing *Csf1r*, which encodes the gene for colony stimulating factor 1 receptor, in the yolk sac, and they are maintained in the fetal liver independent from hematopoietic stem cells [10,11]. On the other hands, BMDMs differentiate and develop from monocytes that originate from bone marrow. It has also been shown that there are at least two kinds of macrophages with different transcriptome profiles that are present in normal human liver [12,13]. Cell surface markers, distribution, and function differ between Kupffer cells and BMDMs [14]. Kupffer cells are radioresistant, show a F4/80^hi^CD11b^−^ phenotype, are present in the midzonal area, and have strong capacity for phagocytosis and reactive oxygen species production [15,16,17]. By contrast, BMDMs are radiosensitive, show F4/80^lo^CD11b^+^, are present in the periportal area, and have higher capacity to produce pro-inflammatory cytokines and promote liver regeneration. Pro-inflammatory signals, including toll-like receptor (TLR) ligands, induce the migration and activation of BMDMs through a mechanism that requires C-C chemokine 2 and its receptor C-C chemokine receptor 2 [18,19]. Interestingly, detailed characterization of mouse liver CD45^+^ leukocytes using CyTOF mass cytometry has identified two kinds of Kupffer cells (F4/80^+^CD11b^+^Ly6C^lo^MHC-II^+^ and F4/80^+^ CD11c^+^CD11b^+^Ly6C^lo^MHC-II^+^) and two kinds of infiltrating monocytes (F4/80^+^Ly6C^+^ and F4/80^-^CD11c^lo^Ly6C^+^) [20]. Additionally, a recent report utilizing single-cell RNA sequencing analysis demonstrates that three kinds of BMDMs can be detected and that their populations are changed by western-type diet feeding [21]. Western diet feeding decreases the expression of *S100a8* and *S100a9* in monocytes, macrophages, and dendritic cells in the liver, which suggests a reduced inflammatory capacity of BMDMs during NAFLD progression, a compensatory mechanism to limit local inflammation. Triggered receptor expressed on myeloid cells 2 (TREM2), a lipid-sensing surface receptor that belongs to the immunoglobulin superfamily, is expressed in myeloid cells, such as microglia, dendritic cells, and osteoclasts, and regulates the development and function of these cells [22]. TREM2 interacts with the adaptor protein DNAX activation protein of 12 kDa, called DAP12, and recruits the Syk tyrosine kinase for signal transmission in cells. TREM2 plays a protective role in mouse models of Alzheimer disease [23], and *TREM2* variants are associated with an increased risk of Alzheimer disease [24,25]. LXR agonist treatment stimulates microglial phagocytosis with the induction of the phagocytic receptors Axl and Mer receptor tyrosine kinase (MerTK) [26]. MerTK^+^/Axl^+^ macrophages surrounding plaques in mouse models of Alzheimer disease express TREM2. The *Mertk* gene is an LXR direct target [27]. TREM2^hi^ macrophages, which reveal enriched *Trem2* mRNA expression, are specifically detected in atherosclerotic aortas in murine models, and TREM2 protein expression is increased in macrophages in human atherosclerotic lesions [28]. *Trem2*-expressing macrophages appear in obese adipose tissue and the genetic deletion of *Trem2* in mice leads to adipocyte hypertrophy, hypercholesterolemia, and glucose intolerance [29]. The *Trem2* gene is highly expressed in Kupffer cells from diet-induced NASH mouse livers and *TREM2* mRNA levels are increased in livers from patients with hepatic steatosis and NASH [30]. The population and function of resident macrophages, including Kupffer cells, and BMDMs, may be dynamically altered by the context of signals and environmental changes to regulate and be involved in metabolic regulation as well as immunity.

### 2.2. Hepatic Stellate Cells (HSCs)

HSCs are mesenchymal cells that are present in the endothelial space of Disse and they are responsible for the promotion of hepatic fibrosis. In quiescent conditions, HSCs contain abundant lipid droplets of retinyl esters. When HSCs are activated by soluble factors including transforming growth factor β and platelet-derived growth factor, these cells transdifferentiate to myofibroblasts, a process that is marked by S100A6 expression, depletion of stored lipids, enhanced proliferation, and fibrogenic action mediated by the induction of α-smooth muscle actin (α-SMA) and collagen α1(I) (Col1a1) [31]. The activity and survival of HSCs are strongly influenced by macrophages, granulocytes, and lymphocytes, and the interaction between HSCs and induced hepatic immune cells contributes to hepatic fibrosis [32].

### 2.3. Liver Sinusoidal Endothelial Cells (LSECs)

LSECs are abundantly present in the liver sinusoid and they play a role in the progression of NAFLD, fibrosis, and hepatocellular carcinoma (HCC) [33]. Unlike endothelial cells in other tissues, LSECs are composed of pores, called fenestrae, and transfer macromolecules, such as chylomicron remnants, between blood and the space of Disse for efficient uptake by hepatocytes. LSECs have strong endocytic activity and they maintain a quiescent HSC phenotype. In pathological conditions, such as chronic liver injury, LSECs are capillarized, lose fenestrae, and allow for HSCs to activate fibrotic action [34].

### 2.4. Natural Killer (NK) Cells and Natural Killer T (NKT) Cells

The innate immune leukocytes NK cells and NKT cells are abundantly present in the rodent liver in physiological conditions. NKT cells are a semi-variant of T cells that express markers of both NK cells (NK1.1) and conventional T cells (αβ-T cell receptor). NKT cells are activated by specific glycolipid antigens, such as α-galactosylceramide (α-GalCer), presented to the MHC class I-like CD1d molecule in antigen-presenting cells [35]. After thymic maturation and development, peripheral NKT cells express an invariant T cell receptor (Vα14-Jα18 and Vβ8.2, Vβ7 or Vβ2 in mouse and Vα24-Jα18 and Vβ11 in human) and produce both the Th1 cytokine interferon-γ and the Th2 cytokine interleukin (IL)-4 [36]. Hepatic invariant NKT (iNKT) cells have multiple functions, such as mediating acute inflammation [37], anti-tumor immunity in cooperation with NK cells [38,39], regeneration [40], and tissue repair [41]. HSECs, which express C-X-C motif ligand 16, are involved in the recruitment of cells positive for C-X-C motif chemokine receptor 6 (CXCR6), particularly NKT cells, in the liver [42]. CXCR6 deletion in mice decreases the intrahepatic numbers of iNKT cells and CD4^+^ cells and enhances diethylnitrosamine-induced hepatocarcinogenesis [43]. Analysis of human samples reveals that *CXCR6* expression is increased in cirrhotic livers but decreased in HCC [43].

## 3. Cholesterol Metabolism and Function of Hepatic Nonparenchymal Cells in Liver Diseases

### 3.1. NAFLD and NASH in Hypercholesterolemic Mice

Cholesterol accumulation in macrophages triggers inflammatory responses that are mediated by TLR signaling and inflammasome activation, and enhances further immune cell recruitment from bone marrow [44]. High cholesterol diet (HCD) or a western-style diet containing high-fat and high-cholesterol diet (HFCD) induces free cholesterol accumulation in hepatocytes and causes NAFLD, NASH, and hepatic fibrosis in mice [45]. Inflammasome activation that is induced by cholesterol crystals is also involved in the progression of NASH [46]. Treatment with statins decreases hepatic free cholesterol content and ameliorates the NASH phenotype in mice that were fed HCD or HFCD [45]. Under lipid-enriched conditions, hepatocytes influence the population and activation of several nonparenchymal cells. Among the immune cells, F4/80^lo^CD11b^+^ BMDMs rather than F4/80^hi^CD11b^-^ resident Kupffer cells are significantly increased in HCD- and HFCD-fed mice, not in high-fat diet (HFD)-fed mice [47]. The relative population of BMDMs to resident Kupffer cells is increased in a cholesterol-dose dependent manner [48]. There are distinct gene expression profiles in resident Kupffer cells and BMDMs in HCD-fed mice. While resident Kupffer cells express genes that are involved in lipid metabolism, tissue repair, and regeneration, BMDMs highly express pro-inflammatory cytokines and chemokines, which promotes further inflammation.

### 3.2. Acute Hepatic Inflammation

HCD- or HFCD-fed mice are susceptible to stimulation with TLR ligands, such as *Escherichia coli*, lipopolysaccharide (LPS), and CpG oligonucleotide, which are associated with strong pro-inflammatory cytokine production, acute liver injury, and increased numbers and activation of F4/80^lo^CD11b^+^ BMDMs [47,49].

### 3.3. Hepatic Fibrosis and Hepatocellular Carcinoma (HCC)

Cholesterol feeding increases free cholesterol accumulation in HSCs as well as hepatocytes. Accumulated cholesterol esters are hydrolyzed to free cholesterol by lysosomal acid lipase, which likely results in cholesterol crystallization [1,50]. Cholesterol crystals are highly proinflammatory and the presence of cholesterol crystals is associated with the development of fibrosing NASH in humans [50]. In a fibrosis model with bile duct ligation (BDL) or carbon tetrachloride (CCl_4_) treatment, HCD feeding enhances the hepatic gene expression of *Acta2* (the gene encoding α-SMA) and *Col1a1* (the gene encoding collagen α1(I)) [51]. α-SMA and collagen α1(I) are specific markers of activated stellate cells [52]. Interestingly, HCD feeding does not affect hepatocyte damage, macrophage infiltration, or pro-inflammatory gene expression induced by BDL or CCl_4_ treatment. In addition, BDL or CCl_4_-induced liver fibrosis is exacerbated in mice lacking acyl-coenzyme A:cholesterol acyltransferase 1, which indicates that free cholesterol accumulation enhances the progression of fibrosis [53]. Increased fibrosis also enhances the development of HCC in long-term CCl_4_-treated, HFCD-fed mice [54]. Thus, excess free cholesterol in HSCs rather than in parenchyma or myeloid cells is strongly involved in the progression of liver fibrosis.

### 3.4. Anti-Tumor Immunity

Hepatic cholesterol accumulation activates not only myeloid cells and HSCs, but also lymphocytes, such as NK cells and NKT cells. In HCD- or HFCD-fed mice, the number of NK cells are increased along with expression of activation marker CD69 [47]. HCD or HFCD feeding increases the sensitivity of mice to α-GalCer treatment, which indicates that NK cells and NKT cells are also activated in HCD-fed mice [49]. Cytotoxic activity against tumor cells is enhanced in hepatic mononuclear cells (MNCs) that are isolated from HFCD-fed mice, and stimulation with LPS or IL-12 improves the survival of HFCD-fed mice in a tumor metastasis model [47].

## 4. Oxysterol Receptors LXRα and LXRβ

### 4.1. Regulation of Lipid Metabolism

LXRα and LXRβ belong to the nuclear receptor superfamily and function as ligand-dependent transcription factors [3]. LXRα is highly expressed in metabolic organs, such as liver (hepatocytes), adipose tissues, kidney, and macrophages, whereas LXRβ is present ubiquitously. LXRs are activated by oxysterols, such as 24(*S*),25-epoxycholesterol, 20(*S*)-hydroxycholesterol, 22(*R*)-hydroxycholesterol, 24(*S*)-hydroxycholesterol, 7α-hydroxycholesterol, 25-hydroxycholesterol, and 27-hydroxycholesterol [55,56,57], and regulate the expression of genes that are involved in lipid metabolism and immunity [3]. Phytosterols and 7-dehydrocholesterol metabolites can act as LXR ligands [58,59]. LXRs preferentially bind to a DNA element that consists of a two-hexanucleotide (AGGTCA or a related sequence) direct repeat motif that is separated by four nucleotides, called the direct repeat 4, as a heterodimer with retinoid X receptor (RXR). Although RXR-LXRα and RXR-LXRβ heterodimers function as permissive heterodimers, in that they can be activated by both LXR and RXR ligands, the physiological role of RXR signaling in the heterodimers remains elusive. LXRs play a central role in the regulation of cholesterol homeostasis. In rodent hepatocytes, oxysterols, which are increased with excess cholesterol, activate LXRα and stimulate the catabolism of cholesterol to bile acid by inducing the expression of the rate-limiting enzyme in the bile acid synthetic pathway, cholesterol 7α-hydroxylase [60]. Hepatocyte LXRα also increases the expression of ATP-binding cassette (ABC) transporter G5 and ABCG8, and it stimulates biliary cholesterol excretion [61]. LXRα-deficient mice that were fed HCD show severe cholesterol accumulation in the liver [60]. In macrophages, LXRs induce the expression of ABCA1 and apolipoprotein E and stimulate reverse cholesterol transport [62,63]. Therefore, LXR agonist treatment suppresses the development of atherosclerosis in a mouse model [64]. Although hepatocyte-specific LXRα knockout mice show a dysregulation of hepatic cholesterol metabolism and accelerated atherogenesis, LXR agonist treatment still has an atheroprotective effect [65]. Bone marrow transplantation of LXRα/β-deficient cells to atherogenic mice increases atherosclerotic lesions in the aorta [66]. Therefore, both liver and macrophage LXRs are required for whole body cholesterol homeostasis and protection against atherosclerosis.

### 4.2. Anti-Inflammatory Function in Immune Cells

LXR agonist treatment suppresses the expression of pro-inflammatory genes, such as inducible nitric oxide synthase and cyclooxygenase-2, as induced by the TLR4 ligand LPS in mouse peritoneal macrophages by inhibiting corepressor protein dissociation from the inflammatory transcription factor nuclear factor κB [67,68,69]. LXR activation also represses TLR ligand-dependent inflammatory effects in an ABCA1-dependent manner, and it directly modulates chromatin accessibility in pro-inflammatory gene enhancers through a *cis*-repressive interaction [70,71]. In atherosclerotic mice, cholesterol accumulation in macrophages induces the production of desmosterol, which acts as an LXR ligand and regulates lipid metabolism and inflammation [72]. Genome-wide analysis shows that LXRα and LXRβ regulate transcription programs in both common and distinct mechanisms [73]. LXRα preferentially regulates the expression of genes that are related to lipid metabolism, apoptosis, inflammation, and immune cell migration, while LXRβ is involved in lymphocyte differentiation. Both LXRs are involved in the regulation of genes for DNA replication and rRNA processing. LXR also regulates immune cell migration by an indirect mechanism through chemotactic signaling in dendritic cells [74]. LXR activation enhances dendritic cell chemotaxis dependent on CCL19 and CCL21 by inducing the expression of CD38, which is involved in leukocyte trafficking and is associated with atherosclerosis. In contrast, tumor cells produce LXR ligands, which suppress CCR7 expression in dendritic cells and attenuate antitumor immunity [75].

CD8^+^ T cells can differentiate into Tc1, Tc2, Tc9, and Tc17 cells under various cytokine conditions, and LXR activation by cholesterol accumulation inhibits the expression of IL-9, which is indispensable for Tc9 cell persistence and antitumor effects [76]. Interestingly, the pharmacological activation of LXR reduces myeloid-derived suppressor cell abundance in murine models and in human cancer patients [77]. LXR activation increases the expression of apolipoprotein E, which induces apoptosis of myeloid-derived suppressor cells, leading to the activation of cytotoxic T cells and effective anti-tumor responses. Furthermore, LXR ligand treatment suppresses Th1 and Th17 polarization and induces Treg differentiation [78]. Thus, LXRs control immune cell function through direct and indirect mechanisms.

## 5. LXR Function in Hepatic Nonparenchymal Cells in Liver Diseases

### 5.1. Expression of LXRs in Liver Nonparenchymal Cells

LXR*α* is highly expressed in rat resident Kupffer cells [79]. In mouse liver, LXR*α* and LXR*β* are both abundantly present in hepatic MNCs, and LXR*α* is preferentially expressed in F4/80^hi^CD11b^lo^ Kupffer cells rather than F4/80^lo^CD11b^+^ BMDMs [80]. Clec4f encodes C-type lectin domain family 4 member F, which is a glycosylated membrane protein that is co-expressed with F4/80 on Kupffer cells and it interacts with *α*-GalCer [81]. A genome-wide study shows that the specific expression of LXR*α* is observed together with Clec4f expression in resident Kupffer cells and that LXR-regulating sequence motifs are enriched in the Kupffer cell-specific enhancer region [82]. The transcription factor zinc finger E box binding homeobox 2, which is known to play a role in epithelial to mesenchymal transition, controls LXR*α* expression to maintain the tissue-specific phenotype in resident Kupffer cells [83]. Thus, LXRs, specifically LXR*α*, are functionally expressed not only in hepatocytes, but also in nonparenchymal cells, including resident Kupffer cells in the physiological condition. The effects of LXR activation or deletion in nonparenchymal cells are summarized (Table 1).

### 5.2. Acute Hepatic Inflammation

Anti-inflammatory effects of LXR agonist are observed in rodent liver Kupffer cells and MNCs as well as in BMDMs or peritoneal macrophages. In cultured Kupffer cells, LXR agonist treatment suppresses tumor necrosis factor (TNF)-α production that is induced by LPS [79,91]. LXR ligand treatment decreases plasma pro-inflammatory cytokine production, ALT levels, and leukocyte infiltration in LPS-challenged rats and mice [79,84,86]. LXR ligand treatment also attenuates LPS-induced transaminitis, TNF-α, and inducible nitric oxide synthase expression in HFD-fed mice [85], and inhibits pro-inflammatory cytokine expression in cultured mouse hepatic MNCs [80]. TLR ligand-stimulated pro-inflammatory gene expression is strongly enhanced in cultured hepatic MNCs from LXRα/β-knockout mice, and LPS administration induces severe hepatocyte damage in LXRα/β-knockout mice [80] (Figure 1). LXRα/β-knockout mice, but not LXRα- or LXRβ-knockout mice, show an increased number of liver F4/80^lo^CD11b^+^ BMDMs and a higher expression of *Ccl2*. CCL2 from hepatocytes recruits myeloid cells expressing its receptor, C-C chemokine receptor 2, to the liver [92]. Therefore, LXRα and LXRβ are both involved in the suppression of acute liver injury by regulating pro-inflammatory gene expression and the recruitment of BMDMs.

### 5.3. Effects of LXR Activation or Absence in NAFLD and NASH

Despite the beneficial effects on cholesterol metabolism and the inhibition of inflammation, LXR agonist stimulates lipogenesis and induces hepatic steatosis by increasing the expression of sterol regulatory element-binding protein-1c, fatty acid synthase, and stearoyl-CoA desaturase [93,94]. Interestingly, the LXR endogenous ligand 27-hydroxycholesterol suppresses HFD-induced steatohepatitis by decreasing leukocyte recruitment and pro-inflammatory gene induction in Kupffer cells without affecting hepatic sterol regulatory element-binding protein-1c expression [95]. A recent study shows that the production of some specific oxysterols, such as 24(*S*)-hydroxycholesterol and 7β-hydroxycholesterol, is increased in the liver of NASH model mice and human patients [5]. Oxysterols might be involved in the pathogenesis of NAFLD/NASH independently from LXR activation, and the role of LXR signaling as oxysterol receptors in the development of NAFLD/NASH remains to be elucidated.

LXRα deletion, not LXRβ deletion, results in increased hepatic cholesterol content, liver weight, and plasma transaminase levels in HCD-fed mice [6,60]. The population of F4/80^+^CD68^+^CD11b^+^ macrophages is increased in LXRα-knockout mice fed HFCD for four weeks, associated with hepatic cholesterol accumulation [87] (Figure 2). LPS challenge in these mice induces more severe hepatic damage. Interestingly, although the expression of the lymphocyte activation marker CD69 is increased in NK cells, the number of iNKT cells is decreased, and the response to the NKT activator α-GalCer is impaired in these mice. Increased cell number and activation of Kupffer cells by lipid overload induces NKT cell death in mouse liver [96]. Choline-deficient diet feeding activates Kupffer cells and increases IL-12 production, which leads to the depletion of NKT cells [97]. The absence of LXRα function induces cholesterol accumulation in hepatocytes, activates Kupffer cells/macrophages, and decreases iNKT cells. A mechanistic understanding of the contribution of LXRs in the regulation of hepatic iNKT cells remains incomplete.

### 5.4. Hepatic Fibrosis

HSCs from LXRα/β-knockout mice contain more lipid droplets and demonstrate an increased expression of several fibrogenic genes, including *Acta2* (the gene encoding α-SMA) and *Col1a1* [88]. Liver fibrosis that is induced by CCl_4_ treatment or methionine/choline-deficient diet is exacerbated, being marked by elevated plasma ALT levels and Col1a1 expression in LXRα/β-knockout mice. Bone marrow transplantation of wild-type cells into LXR-null mice does not affect the fibrosis score, which indicates that the antifibrogenic effect of LXRs is not due to hematopoietic cells, such as BMDMs and lymphocytes. LXRα/β-deficient HSCs store more cholesterol and retinyl esters and they have more retinoic acid-mediated profibrogenic action [98]. CCl_4_-treated LXRα-knockout mice also show the activation of LSECs, such as elevation of CD34 expression and capillarization [99]. Conversely, LXR ligand treatment suppresses capillarization through a mechanism that is mediated by Hedgehog signaling [89]. Genome-wide association studies have revealed that the patatin-like phospholipase domain-containing protein 3 (PNPLA3) I148M variant is associated with a higher risk of hepatic steatosis, fibrosis, and HCC [100]. PNPLA3 has lipase activity towards triglycerides in hepatocytes and retinyl esters in HSCs. Human HSCs expressing I148M PNPLA3 have decreased peroxisome proliferator-activated receptor γ activity, which leads to impaired LXR signaling and cholesterol accumulation [101]. LXRs in HSCs and LSECs play a role in the regulation of liver fibrosis.

### 5.5. Wilson Disease

Wilson disease is caused by a genetic mutation of the ATP-dependent copper transporter protein ATP7B and it is characterized by hepatic copper accumulation and hepatitis. Microarray analysis shows that the expression of LXR-regulated lipid metabolism genes is impaired in the liver of patients with Wilson disease [90]. Protein levels of RXR, not LXRα or LXRβ, are decreased in the liver of Atp7b-deficient mice, a model of Wilson disease. LXR ligand treatment of Atp7b-deficient mice does not change hepatic copper levels, but it reduces the expression of pro-inflammatory and profibrogenic genes and attenuates liver injury.

## 6. Concluding Remarks

LXRs are important regulators of cholesterol homeostasis and they have been investigated as promising therapeutic targets in the treatment of hypercholesterolemia and atherosclerosis. Impaired hepatic cholesterol metabolism is strongly associated with the pathogenesis and progression of HCD- and HFCD-induced liver diseases. LXR ligands are also candidate treatments for western diet-related liver diseases, such as NASH. LXRs are expressed not only in hepatocyte, but also in nonparenchymal cells, such as Kupffer cells, BMDMs, HSCs, and LSECs, and they play important roles in the regulation of inflammation, immune cell population, and activation through mechanisms that are related to and independent of their metabolic action. While many synthetic LXR agonists have been developed for the treatment of hypercholesterolemia and cancer, their clinical application is limited by hypertriglyceridemia and other adverse effects on the central nervous system [102,103]. Further studies are needed to develop cell- or function-selective ligands that target hepatic leukocytes, HSCs, and LSECs.

## Figures and Tables

**Figure 1 ijms-20-05045-f001:**
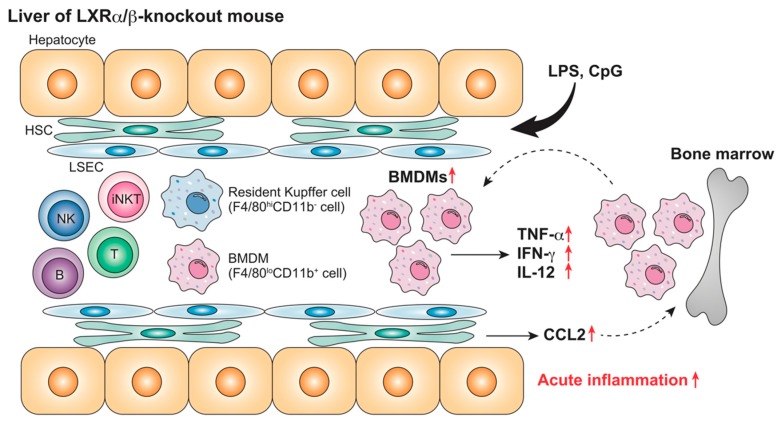
Acute hepatic inflammation in LXRα/β-knockout mice. Lipopolysaccharide (LPS) administration stimulates inflammatory cytokine production and recruitment of F4/80^lo^CD11b^+^ bone marrow-derived macrophages (BMDMs) in the liver of LXRα/β-knockout mice more effectively than in wild-type mice [80]. LXRs regulate acute hepatic inflammatory responses. ↑, increase.

**Figure 2 ijms-20-05045-f002:**
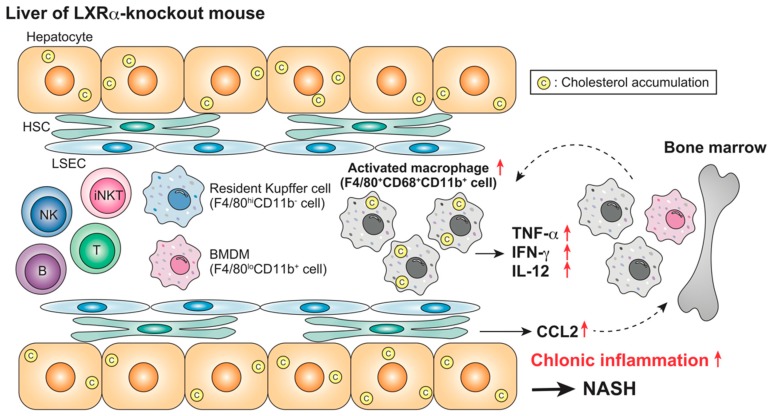
Nonalcoholic steatohepatitis (NASH) in high-cholesterol diet (HFCD)-fed LXRα-knockout mice. HFCD feeding induces severe steatohepatitis in LXRα-knockout mice, associated with an increase in CD68^+^CD11b^+^ Kupffer cells/macrophages and inflammatory responses [87]. LXRα plays a protective role against NASH by regulating cholesterol metabolism and immune responses. ↑, increase.

**Table 1 ijms-20-05045-t001:** Effects of liver X receptor (LXR) activation or deletion in hepatic nonparenchymal cells in animal models of liver disease.

Animal	LXR	Acute or Chronic	Treatment	Target Cells	Findings	Ref
Mouse (WT, LXRα/β-KO)	Absence		No treatment	F4/80^lo^CD11b^+^ Kupffer cells/macrophages	F4/80^lo^CD11b^+^ cells↑ Inflammatory gene expression in hepatic MNCs↑	[80]
Rat (WT)	Activation	Acute	GW3965 (0.1 or 0.3 mg/kg) LPS (5 mg/kg)/ Peptidoglycan (1 mg/kg) for 1–6 h	Kupffer cells/macrophages	Plasma TNF-α, PGE2, ALT, bilirubin ↓ Hepatocyte injury ↓	[79,84]
Mouse (WT)	Activation	Acute	HFD (8 weeks), T0901317 (30 mg/kg for 7 days) LPS (50 μg/mouse)	Kupffer cells/macrophages	Plasma AST/ALT ↓ Hepatic TNF-α, iNOS expression ↓ Hepatic PI3K, NF-κB, MAPK signaling ↓	[85]
Mouse (WT)	Activation	Acute	T0901317 (50 mg/kg for 12 h) LPS (10 mg/kg for 12 h)	Kupffer cells/macrophages	Leukocyte infiltration ↓ Plasma AST/ALT ↓ Plasma, supernatant of Kupffer cells TNF-α, IFN-β, IL-1β ↓, IL-10 ↑	[86]
Mouse (WT, LXRα-KO)	Absence	Chronic (NASH)	HFCD (4 weeks)	Kupffer cells/macrophages iNKT cells	F4/80^+^CD68^+^CD11b^+^ cells ↑ iNKT cells ↓ Plasma AST/ALT ↑	[87]
Mouse (WT, LXRα-KO)	Absence	Chronic (NASH)	HFCD (4 weeks) LPS (2.5 mg/kg, 1–6 h)	Kupffer cells/macrophages	Leukocyte accumulation ↑ Plasma TNF-α, IFN-γ, IL-12, CCL2, AST/ALT ↑	[87]
Mouse (WT, LXRα-KO)	Absence	Chronic (NASH)	HFCD (4 weeks) α-GalCer, 0.1 mg/kg, 1–72 h)	iNKT cells	Plasma IL-4, IFN-γ ↓	[87]
Mouse (WT, LXRα-KO)	Absence	Chronic (NASH)	HFCD (4 weeks) Con-A, 12.5 mg/kg, 1–72 h)	iNKT cells	Plasma AST/ALT, bilirubin ↓ Hepatocyte necrosis ↓	[87]
Mouse (WT, LXRα/β-KO)	Absence	Chronic (Fibrosis)	CCl_4_ (10%, *v*/*v*, twice per week for 4 weeks)	HSCs	Plasma ALT ↑ *Col1a1* expression ↑ Fibrosis ↑	[88]
Mouse (WT, LXRα/β-KO)	Absence	Chronic (Fibrosis)	Methionine/choline-deficient diet (1 month)	HSCs	*Col1a1* expression ↑ Fibrosis ↑	[88]
Mouse (WT)	Activation	Chronic (Fibrosis)	T0901317 (50 mg/kg, daily for 1 week) CCl_4_ (10 %, *v*/*v*, twice per week for 1 month)	LSECs	CD31^+^ LSEC capillarization ↓ Hepatic F4/80^+^ cells ↓ Fibrotic score ↓	[89]
Mouse (WT, LXRα-KO)	Absence	Chronic (Fibrosis)	CCl_4_ (10%, *v*/*v*, twice per week for 1 month)	LSECs	CD31^+^ LSEC capillarization ↑ Hepatic F4/80^+^ cells ↑ Fibrotic score ↑	[89]
Mouse (*Atp7b*-KO)	Activation	Chronic (Wilson disease)	T0901317 (50 mg/kg, thrice per week for 8 weeks)	Kupffer cells/macrophages HSCs	Hepatic inflammatory or fibrotic genes ↓ Plasma AST/ALT ↓	[90]

Arrows indicate as follows: ↑, increase; ↓, decrease.

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
