# Peer review of "Liver X Receptors Regulate Cholesterol Metabolism and Immunity in Hepatic Nonparenchymal Cells"

_ijms, 2019, doi:10.3390/ijms20205045_

Round 1

Reviewer 1 Report

This article reviews the roles of LXRs in hepatic nonparenchymal cell functions and their putative involvement in the pathogenesis of hepatic diseases.

This article well written and easy to read. The figures are clear and adequately support the text.

I only have minor comment:

Lines 84 to 95: TREM2 seems to have a central metabolic role in the liver. The normal function of TREM2 protein should be detailed. However, its putative relationships with LXRs could be discussed.

Line 111: please define HSP abreviation.

Line 154: Although presented later as fibrogenic genes (line 295), the authors should briefly introduce the participation of α-SMA and Col1a1 genes to hepatic fibrosis when cited for the first time.

Some orthograph errors are present, including:

Line 35: replace NR1H by NR1I.

Line 85: replace moue by mouse.

Line 303: replace genobe by genome.

Line 305: replace associated by associated.

Line 308: replace aginaling by signaling.

Author Response

Comment by Reviewer #1

This article reviews the roles of LXRs in hepatic nonparenchymal cell functions and their putative involvement in the pathogenesis of hepatic diseases.

This article well written and easy to read. The figures are clear and adequately support the text.

I only have minor comment:

Comment #1

Lines 84 to 95: TREM2 seems to have a central metabolic role in the liver. The normal function of TREM2 protein should be detailed. However, its putative relationships with LXRs could be discussed.

Response

We thank the reviewer for the favorable comments.

We added explanation of TREM2 as follows.

“Triggered receptor expressed on myeloid cells 2 (TREM2), a lipid-sensing surface receptor that belongs to the immunoglobulin superfamily, is expressed in myeloid cells, such as microglia, dendritic cells and osteoclasts, and regulate the development and function of these cells [20]. TREM2 interacts with the adaptor protein DNAX activation protein of 12 kDa, called DAP12, and recruits the Syk tyrosine kinase for signal transmission in cells. TREM2 plays a protective role in mouse models of Alzheimer disease [21], and TREM2 variants are associated with an increased risk of Alzheimer disease [22,23]. LXR agonist treatment stimulates microglial phagocytosis with induction of the phagocytic receptors Axl and Mer receptor tyrosine kinase (MerTK) [24]. MerTK+/Axl+ macrophages surrounding plaques in mouse models of Alzheimer disease express TREM2. The Mertk gene is an LXR direct target [25]” (lines 86-95).

Comment #2

Line 111: please define HSP abreviation.

Response

We made a typo and corrected it to “HSC” (line 119).

Comment #3

Line 154: Although presented later as fibrogenic genes (line 295), the authors should briefly introduce the participation of α-SMA and Col1a1 genes to hepatic fibrosis when cited for the first time.

Response

We added explanation of α-SMA and Col1a1 as follows.

“α-SMA and collagen α1(I) (encoded by Col1a1) are specific markers of activated stellate cells [50]” (lines 165-166).

Comment #4

Some orthograph errors are present, including:

Line 35: replace NR1H by NR1I.

Response

We thank the reviewer for indicating our careless typos.

We corrected it to “NR1I” (line 35).

Comment #5

Line 85: replace moue by mouse.

Response

We corrected it to “mouse” (line 90).

Comment #6

Line 303: replace genobe by genome.

Response

We corrected it to “Genome” (line 316).

Comment #7

Line 305: replace associated by associated.

Response

We corrected it to “associated” (line 318).

Comment #8

Line 308: replace aginaling by signaling.

Response

We corrected it to “signaling” (line 320).

Comment #9

In addition, some words were replaced to the corrects as follows:

Line 211: CCL7 to “CCR7”

Response

We corrected it to “CCR7” (line 225).

Comment #10

Line 306: heptatocyte to “hepatocytes”

Response

We corrected it to “hepatocytes” (line 319).

Comment #11

Line 306: expressiong to “expressing

Response

We corrected it to “expressing” (line 320).

Comment #12

Line 307: decreaed to “decreased”

Response

We corrected it to “decreased” (line 320).

Reviewer 2 Report

The manuscript 'Liver X Receptors Regulate Cholesterol Metabolism and Immunity in Hepatic Nonparenchymal Cells' by Endo-Umeda et al. comprehensively reviews up to date knowledge on the role of LXRs connecting cholesterol metabolism and inflammatory/immune-regulation focussing on non-parenchymal liver cells.

The review is overall well written, contents are presented in logical sequence and are supported by illustrations and a table giving an overview on results obtained from animal studies.

Suggestions:

Abstract: 9: Excess cholesterol intake can be understood as dietary intake or increased cellular cholesterol – Please clarify.

13: (LXR)a not LXR a

14: Cholesterol efflux belongs to the reverse cholesterol transport pathway, may be changed.

A list of abbreviations can be included if supported by the journal style.

Introduction: 32: LXR(a)

33: in hepatocytes is not accurate, since LXRs are expressed in many other cells (as described later in the text also). Please change.

49: LXR agonists may be explained in short

152: Can authors explain how (and why) free (or better unesterified) cholesterol accumulates in cells?

174: the authors mean 24(s),25-epoxycholesterol. The most relevant endogenous LXR ligands should all be mentioned here – including 27OH-C and 24(S)OH-C.

Several subheadings of individual chapters are identical (e.g. 3.1. and 5.3.) which may be misleading. Subheadings should describe the content of the paragraphs more accurately.

Also, some headings are abbreviations which should rather be spelled out (e.g. 2.2. HSCs…).

Table: the layout/format can be improved to increase clarity on which words in the rows underneath belong together. Some inconsistencies about when to use capital letters (e.g. ALT, Bilirubin -> bilirubin). Mpk or mg/kg?

1st row: no treatment appears twice.

Figure legends: the citations are missing in the legends (70 and 77 for Fig.1 and 2, respectively).

Typos: 73: and promote of liver regeneration 303: Genome-

159: free cholesterol not cholesterols

308: signaling

                311: caused by a genetic mutation

Author Response

Comment by Reviewer #2

The manuscript 'Liver X Receptors Regulate Cholesterol Metabolism and Immunity in Hepatic Nonparenchymal Cells' by Endo-Umeda et al. comprehensively reviews up to date knowledge on the role of LXRs connecting cholesterol metabolism and inflammatory/immune- regulation focussing on non-parenchymal liver cells.

The review is overall well written, contents are presented in logical sequence and are supported by illustrations and a table giving an overview on results obtained from animal studies.

Comment #1

Abstract: 9: Excess cholesterol intake can be understood as dietary intake or increased cellular cholesterol – Please clarify.

Response

We thank the reviewer for the favorable comments.

It’s “dietary”, which was added (line 9).

Comment #2

13: (LXR)a not LXR a

Response

We changed “liver X receptor (LXR) a” to “liver X receptor a (LXRa)” (line 13).

Comment #3

14: Cholesterol efflux belongs to the reverse cholesterol transport pathway, may be changed.

Response

We changed “efflux” to “excretion” (line 15).

Comment #4

A list of abbreviations can be included if supported by the journal style.

Introduction: 32: LXR(a)

Response

We changed “liver X receptor (LXR) a” to “liver X receptor a (LXRa)” (line 32).

Comment #5

33: in hepatocytes is not accurate, since LXRs are expressed in many other cells (as described later in the text also). Please change.

Response

We deleted “in hepatocytes” (line 33).

Comment #6

49: LXR agonists may be explained in short

Response

We added explanation of the LXR agonist development as follows.

“Although synthetic LXR agonists have been developed, their clinical application is limited by adverse effects, such as hypertriglyceridemia and neuropsychiatric symptoms [94,95]. Function-selective LXR agonists may have promising therapeutic potential for liver diseases by regulating lipid metabolism and immune responses” (lines 49-51)

Comment #7

152: Can authors explain how (and why) free (or better unesterified) cholesterol accumulates in cells?

Response

We added explanation as follows.

“Accumulated cholesterol esters are hydrolyzed to free cholesterol by lysosomal acid lipase, likely resulting in cholesterol crystallization [1,48]. Cholesterol crystals are highly proinflammatory and the presence of cholesterol crystals is associated with the development of fibrosing NASH in humans [48]” (lines 161-164).

Comment #8

174: the authors mean 24(s),25-epoxycholesterol. The most relevant endogenous LXR ligands should all be mentioned here – including 27OH-C and 24(S)OH-C.

Response

We added such oxysterol ligands as follows.

“LXRs are activated by oxysterols, such as 24(S),25-epoxycholesterol, 20(S)-hydroxycholesterol, 22(R)-hydroxycholesterol, 24(S)-hydroxycholesterol, 7a-hydroxycholesterol, 25-hydroxycholesterol and 27-hydroxycholesterol [53-55] (lines 186-189).

Comment #9

Several subheadings of individual chapters are identical (e.g. 3.1. and 5.3.) which may be misleading. Subheadings should describe the content of the paragraphs more accurately.

Response

We changed 3.1 and 5.3 subheading as follows.

3.1. NAFLD and NASH in Hypercholesterolemic Mice (line 138)

5.3. Effects of LXR activation or absence in NAFLD and NASH (line 275)

Comment #10

Also, some headings are abbreviations which should rather be spelled out (e.g. 2.2. HSCs...).

Response

We changed the subheading as follows.

2.1. Two Types of Macrophages: Resident Kupffer Cells and Bone Marrow-Derived Macrophages (BMDMs) (line 63)

2.2. Hepatic Stellate Cells (HSCs) (line 104)

2.3. Liver Sinusoidal Endothelial Cells (LSECs) (line 114)

2.4. Natural Killer (NK) Cells and Natural Killer T (NKT) Cells (line 122)

3.3. Hepatic Fibrosis and Hepatocellular Carcinoma (HCC) (line 159)

Comment #11

Table: the layout/format can be improved to increase clarity on which words in the rows underneath belong together. Some inconsistencies about when to use capital letters (e.g. ALT, Bilirubin -> bilirubin). Mpk or mg/kg?

Response

In Table 1, we changed the layout to make a reader to understand easily, made corrections as indicated, and added detailed information of treatment conditions.

Comment #12

1st row: no treatment appears twice.

Response

We deleted “no treatment” in “Acute or Chronic” column.

Comment #13

Figure legends: the citations are missing in the legends (70 and 77 for Fig.1 and 2, respectively).

Response

We added the citations in figure legends.

Comment #14

Typos:

73: and promote of liver regeneration

Response

We thank the reviewer for indicating our careless typos.

We corrected it to “promote liver regeneration” (line 75).

Comment #15

303: Genome-

Response

We corrected it to “Genome” (line 316).

Comment #16

159: free cholesterol not cholesterols

Response

We corrected it to “cholesterol” (line 171).

Comment #17

308: signaling

Response

We corrected it to “signaling” (line 320).

Comment #18

311: caused by a genetic mutation

Response

We corrected it to “caused by a genetic mutation” (line 324).